# Imitation Learning from Visual Data with Multiple Intentions

**Aviv Tamar**[*,1]**, Khashayar Rohanimanesh**[*,2]**, Yinlam Chow**[2]**, Chris Vigorito**[2]**, Ben Goodrich**[2]**,
**Michael Kahane**[2]**, and Derik Pridmore**[2]

[1]EECS Department, UC Berkeley
[2]Osaro Inc.
[*]Equal contribution
avivt@berkeley.edu,    {khash,ychow,chris,ben,mk,derik}@osaro.com

## Abstract

Recent advances in learning from demonstrations (LfD) with deep neural networks have enabled learning complex robot skills that involve high dimensional perception such as raw image inputs. LfD algorithms generally assume learning from single task demonstrations. In practice, however, it is more efficient for a teacher to demonstrate a multitude of tasks without careful task set up, labeling, and engineering. Unfortunately in such cases, traditional imitation learning techniques fail to represent the multi-modal nature of the data, and often result in sub-optimal behavior. In this paper we present an LfD approach for learning multiple modes of behavior from visual data. Our approach is based on a stochastic deep neural network (SNN), which represents the underlying intention in the demonstration as a stochastic activation in the network. We present an efficient algorithm for training SNNs, and for learning with vision inputs, we also propose an architecture that associates the intention with a stochastic attention module. Furthermore, we demonstrate our method on real robot visual object reaching tasks, and show that it can reliably learn the multiple behavior modes in the demonstration data. Video results are available at https://vimeo.com/240212286/fd401241b9.

## 1 Introduction

A key problem in robotic control is to simplify the problem of programming a complex behavior. Traditional control engineering approaches, which rely on accurate manual modeling of the system environment, are very challenging to apply in modern robotic applications where most sensory inputs come from images and other high-dimensional signals such as tactile feedback.

In contrast, imitation learning, or learning from demonstration (LfD) approaches (Schaal et al., 2003) aim to directly learn a control policy from mentor or expert demonstrations. The key advantages of LfD are simplicity and data-efficiency, and indeed, LfD has been successfully used for learning complex robot skills such as locomotion (Schaal et al., 2005), driving (Pomerleau, 1989; Ross et al., 2011), flying (Abbeel & Ng, 2004), and manipulation (Mülling et al., 2013; Chebotar et al., 2016; Pastor et al., 2009). Recently, advances in deep representation learning (Goodfellow et al., 2016) have facilitated LfD methods with high dimensional perception, such as mapping raw images directly to controls (Giusti et al., 2016). These advances are capable of learning generalizable skills (Levine et al., 2015), and offer a promising approach for modern industrial challenges such as pick and place tasks (Correll et al., 2016).

One challenge in LfD, however, is learning different modes of the same task. For example, consider learning to pick up an object from a pile. The demonstrator can choose to pick up a different object each time, yet we expect LfD to understand that these are similar demonstrations of the same pick-

up skill, only *with a different intention in mind*. Moreover, we want the learned robot behavior to display a similar multi-modal[1] nature.

Standard approaches for LfD with image inputs, such as learning with deep neural networks (NNs) (Pomerleau, 1989; Giusti et al., 2016; Levine et al., 2015), are not suitable for learning multi-modal behaviors. In their essence, NNs learn a deterministic mapping from observation to control, which cannot represent the inherently multi-modal latent intention in the demonstrations. In practice, this manifests as an 'averaging' of the different modes in the data (Bishop, 1994), leading to an undesirable policy.

A straightforward approach for tackling the multi-modal problem in LfD is to add a label for each mode in the data. Thus, in the pick-up task above, the demonstrator would also explicitly specify the object she intends to pick-up beforehand. Such an approach has several practical shortcomings: it requires the demonstrator to record more data, and requires the possible intentions to be specified in advance, making it difficult to use the same recorded data for different tasks. More importantly, such a solution is conceptually flawed – it solves an algorithmic challenge by placing additional burden on the client.

In this work, we propose an approach for LfD with multi-modal demonstrations that does not require any additional data labels. Our method is based on a stochastic neural network model, which represents the latent intention as a random activation in the network. We propose a novel and efficient learning algorithm for training stochastic networks, and present a network architecture suitable for LfD with raw image inputs, where the intention takes the form of a stochastic attention over features in the image.

We show that our method can reliably reproduce behavior with multiple intentions in real-robot object reaching tasks. Moreover, in scenarios where multiple intentions exist in the demonstration data, the stochastic neural networks perform better than their deterministic counterparts.

## 2 RELATED WORK

In this work we focus on a direct imitation learning approach, known as behavioral cloning (Pomerleau, 1989). This approach does not require any model of the task dynamics, and does not require additional queries of the mentor or robot execution rollouts beyond the collected demonstrations (though such can be used to improve performance (Ross et al., 2011)). An alternative approach is inverse reinforcement learning (IRL) (Ng & Russell, 2000; Abbeel & Ng, 2004; Ziebart et al., 2008), where a reward model that explains the demonstrated behavior is sought. Recently, model-free IRL approaches that can learn complex behavior policies from high-dimensional data were proposed (Finn et al., 2016; Ho & Ermon, 2016). These approaches, however, rely on taking additional policy rollouts as *a fundamental step of the method*, which, in realistic robot applications, requires substantial resources.

Multi-task IRL learns from unlabeled demonstrations generated by varying intentions or objectives (Babes et al., 2011; Dimitrakakis & Rothkopf, 2012). Dimitrakakis & Rothkopf (2012) propose a Bayesian approach for inferring the intention of an agent performing a series of tasks in a dynamic environment. Babes et al. (2011) propose an EM for clustering the unlabeled demonstrations and then application of IRL for inferring the intention of a given cluster. Both approaches have been shown promising results on relatively simple low dimensional problems. Several recent works on multi-task IRL (Hausman et al., 2017; Wang et al., 2017; Li et al., 2017) extended the generative adversarial imitation learning (GAIL) algorithm (Ho & Ermon, 2016) to high dimensional multi-modal demonstrations. Our approach, in comparison, does not require taking additional robot rollouts.

Recently, Rahmatizadeh et al. (2017) proposed a method for learning multi-modal policies from raw visual inputs, using mixture density networks (Bishop, 1994) for generating outputs from a mixture of Gaussians distribution. Their training method requires labeling each task with a specific signal. While the modes, or intentions in our work can be seen as different tasks, we do not require any labeling of the intention in the demonstrations.

To our knowledge, this is the first LfD approach that can handle multiple modes in the demonstrations and: (1) does not require additional robot rollouts, (2) does not require a label for the mode, and (3) can work with raw image inputs.

---

[1]In this paper, multi-modal refers to a distribution that contains multiple modes. In robotics literature, multi-modal can also refer to policies that act on different input modalities such as vision and sound. We emphasize that this is not the setting in this paper.

The stochastic neural network model we use here is related to recently proposed generative models such as (conditional) variational autoencoders (VAEs) (Kingma & Welling, 2014; Sohn et al., 2015), and generative adversarial nets (GANs) (Goodfellow et al., 2014; Mirza & Osindero, 2014). We opted for stochastic neural networks since GANs are known to have problems learning multi-modal distributions (Arora & Zhang, 2017), and conditional VAEs (Pillai & Leonard, 2017) require training an additional encoding network, which proved to be difficult in our experimental domain.

Very recently, in the context of multi-modal video prediction, Fragkiadaki et al. (2017) proposed the K-best loss for training stochastic neural networks, which is similar to our proposed training algorithm. In that work, stochastic neural networks with K-best loss were also shown to outperform conditional VAEs on some domains. Our contribution, compared to the work of Fragkiadaki et al. (2017), is providing a formal mathematical treatment of this method, proposing optimistic sampling which significantly improves its performance, and showing its importance in a real world robotic imitation learning domain.

## 3 PRELIMINARIES

We first introduce some preliminary concepts for presenting our methods, and then present our problem formulation of imitation learning with multi-modal behaviors in expert demonstrations.

**Learning from Demonstration**    To explicitly formulate the problem of imitation learning, let $\mathcal{X}$ and $\mathcal{U}$ denote the observation and action spaces of the robot, and let $x_t \in \mathcal{X}$ and $u_t \in \mathcal{U}$ denote an observation and a control command for the robot at time $t$. Given a data-set $\mathcal{D}$ of $N$ trajectories $\mathcal{T}^i$ with length $T$ (for simplifying notations we drop the subscript $i$ in $T_i$), where a demonstrated task is recorded in the form of sequential pairs of observations and actions $\mathcal{T}^i = \left\{ \langle x_1^i, u_1^i \rangle, \ldots, \langle x_T^i, u_T^i \rangle \right\}_{i=1}^N$, LfD aims to learn a policy $P : \mathcal{X} \to \mathbb{P}(\mathcal{U})$ that is parametrized by feature weight vector $\theta \in \Theta$, such that it reliably performs the task. Here $\mathbb{P}(\mathcal{U})$ represents the space of probability distributions defined on the action space $\mathcal{U}$. Since each observation is associated with an action label, the imitation learning policy can be found by solving the maximum-likelihood (ML) objective: $\theta^* \in \arg\max_{\theta \in \Theta} \frac{1}{N} \sum_{i=1}^N \log P(u_{1:T}^i | x_{1:T}^i, \theta)$, where we abbreviated the sequence of actions as $u_{1:T} \doteq u_1, \ldots, u_T$, and the sequence of observations as $x_{1:T} \doteq x_1, \ldots, x_T$. This objective function is the empirical average of the conditional log likelihood, which is a consistent estimator of the expected conditional log likelihood: $\mathbb{E}\left[\log P(u_{1:T} | x_{1:T}, \theta)\right]$. If, for example, the policy is Gaussian with parametrized mean vector $f(x; \theta)$ and an identity co-variance matrix, then the above supervised learning problem is equivalent to an $\ell_2$ regression problem with objective function $\frac{1}{N} \sum_{i=1}^N \sum_{t=1}^T \|u_t^i - f(x_t^i; \theta)\|_2^2$.

**Stochastic Neural Networks**    Multilayer perceptrons (MLPs) are general purpose function approximators for nonlinear regression in feedforward neural networks (NNs) (Goodfellow et al., 2016). Parametrized by the NN weights $\theta$, the output of the MLP, $f(x; \theta)$, is often interpreted as the sufficient statistics of the conditional probability $P(u|x; \theta)$, if the conditional probability belongs to the exponential family (conditioned on the input $x$). For example, if $P(u|x; \theta)$ is parametrized as an isotropic Gaussian distribution, it can be represented by $\mathcal{N}(u|f(x; \theta), I)$. The parameters $\theta$ are typically learned by maximizing the expected log likelihood function. However, since the MLP activation functions are all deterministic, by nature the model $P(u|x; \theta)$ is a unimodal distribution.

For many structured prediction problems, we are interested in a conditional distribution that is multi-modal. To satisfy the multi-modality requirement, a common approach is to make the hidden variables in the NN stochastic. Sigmoid belief nets (SBNs) (Neal, 1992) are an early example of this idea, using binary stochastic hidden variables. However, inference in SBNs is generally intractable, and costly to compute in practice. Recently, Tang & Salakhutdinov (2013) introduced the stochastic feedforward neural network (SNN) for modeling the multi-modal conditional distribution $P(u|x; \theta)$. Unlike SBNs, SNNs add to the deterministic NN latent features a stochastic latent variable $z$, and decompose the conditional distribution as: $P(u|x; \theta) = \sum_z P(u|x, z; \theta)P(z)$. It is also assumed that $P(u|x, z, \theta)$ and $P(z)$ can be easily computed, for example, as in (Tang & Salakhutdinov, 2013), where $z$ is represented by Bernoulli random variable nodes in the network. For learning the parameters $\theta$, Tang & Salakhutdinov (2013) proposed a generalized EM algorithm, where importance sampling is used in the E-step, and error back-propagation is used in the M-step.

**Problem Formulation**    In this work, we consider an imitation learning setting, where the mentor demonstrations of a particular task consist of multiple behaviors. In particular, we assume that the

demonstrator can perform the task in several different strategies, which we term as *intentions*. As a concrete example, consider the task of picking up an object from a pile of different objects in the scene (see Figure 1(a) for an example). In this case, the data-set of mentor demonstrations consists of a list of trajectories, where the target object in each trajectory is inherently decided by the demonstrator, and not explicitly labeled. Our goal is to learn a stochastic policy that accurately mimics the mentor's policy, and accurately displays the multiple intentions demonstrated in the data.

## 4 MULTI MODAL LFD WITH SNNS

In this section, we first present our LfD formulation based on the SNN model for learning multiple intentions, and then propose a sampling based algorithm for learning the SNN parameters. We then present a particular SNN architecture that is suitable for vision-based inputs, where the stochastic intention takes the form of an attention over image features.

### 4.1 THE SNN FORMULATION

We model the intention of the demonstrator using a random vector $z \in \mathbb{R}^M$ with probability distribution $\mathcal{P}(z)$. For example, $\mathcal{P}(z)$ could be a unit normal distribution $\mathcal{N}(0, \mathcal{I})$, or a vector of independent multinomial probabilities. Here we assume that throughout a single trajectory, the intention does not change, and the intention is independent of the observations[2]. Therefore, the conditional data likelihood is obtained by marginalizing out the random variable of intention: $P(u_{1:T}|x_{1:T}; \theta) = \sum_z P(u_{1:T}|x_{1:T}, z; \theta)P(z)$. SNNs can be viewed as directed graphical models (see Figure 4 in the appendix for a diagram) where at each time $t \in \{1, \ldots, T\}$, the generative process starts from an observation $x_t$, combines with a latent intention $z$, which is the same throughout the trajectory, and then generates an action $u_t$. We also make the standard assumption that, given the intention $z$ at each trajectory, the demonstrator policy is memory-less (a.k.a. Markov), which implies the following equality: $P(u_{1:T}|x_{1:T}, z; \theta) = \prod_{t=1}^{T} P(u_t|x_t, z; \theta)$.

Given an intention $z$, we model the action probability as $\log P(u|z, x; \theta) \propto -d(f(x, z; \theta), u)$, where $f$ is a deterministic NN that takes as input both the observation $x$ and the intention $z$, and $d$ is some distance metric. One immediate example is when $d(a, b) = \|a - b\|^2$, one obtains $P(u|z, x; \theta)$ as a normal distribution $\mathcal{N}(u|f(x, z; \theta), \sigma^2)$ for some MLP mean predictor $f(x, z; \theta)$ and constant variance term $\sigma^2$. In our experiments, we found the $\ell_1$ distance function to work well. Note that when the intention variable $z$ is fixed, the output action follows a unimodal distribution. However, since $z$ is a random vector that is input to a nonlinear NN computation, the distribution of $f(x, z; \theta)$, and thereby the output distribution $P(u_{1:T}|x_{1:T}; \theta)$, can take a multi-modal form.

### 4.2 THE MONTE CARLO LEARNING ALGORITHM

We first describe a basic Monte Carlo (MC) sampling algorithm for learning the parameters $\theta \in \Theta$ of the SNN model in Section 4.1. Let $z_1, \ldots, z_N$ denote $N$ samples of $z$, where $z_i \sim P(z)$. For each given parameter $\theta$, sequence of observations $x_{1:T}$, and sequence of actions $u_{1:T}$, let $r(z; x_{1:T}, u_{1:T}, \theta) = P(u_{1:T}|x_{1:T}, z; \theta)$ be the *reward function*, which associates an intention with the data likelihood given it. The reason we use this terminology is to later connect the likelihood maximization problem with risk-sensitive optimization concepts that will be key in our approach. A Monte Carlo approximation of the likelihood is given by,

$$P(u_{1:T}|x_{1:T}; \theta) = \mathbb{E}_{z \sim P}[r(z; x_{1:T}, u_{1:T}, \theta)] \approx \frac{1}{N} \sum_{i=1}^{N} r(z_i; x_{1:T}, u_{1:T}, \theta). \tag{1}$$

A direct approach for computing $\theta$ would be to directly maximize $\frac{1}{N} \sum_{i=1}^{N} \log r(z_i; x_{1:T}, u_{1:T}, \theta)$, which is a MC estimate of $\mathbb{E}_{z \sim P}[\log r(z; x_{1:T}, u_{1:T}, \theta)]$, with gradient-based optimization. By Jensen's inequality, the above term is a lower bound of the data log likelihood $\log P(u_{1:T}|x_{1:T}; \theta)$, corresponding to a maximum-likelihood approach. While the estimator of the gradient is unbiased and consistent to $\mathbb{E}_{z \sim P}[\nabla_\theta \log r(z; x_{1:T}, u_{1:T}, \theta)]$, in practice, such an approach suffers from extremely high variance (with respect to intention $z$). To justify this observation, consider the sum

---

[2]By assuming an input-independent intention, our generation model $P(u|x)$ is similar to the generation model in a conditional VAE (Sohn et al., 2015), or conditional GAN (Mirza & Osindero, 2014), although our training method is different. We found the input-independent model to be sufficient for our experiments. Extending our method to observation-dependent intention is possible, along the lines of (Tang & Salakhutdinov, 2013), and deferred to future work.

of probabilities in (1). Since each sampled intention $z_i$ is given an equal weight in explaining the observed sequence of actions, even sampled intentions that are very different from the ones that generated the data (i.e., have high cost) are expected to produce a high likelihood.

### 4.3 INTENTION-DRIVEN LEARNING IN SNNS

To reduce variance in training SNNs, in this section we introduce a sampling strategy whose gradient updates focuses only on the *most correct* underlying intentions – the intentions that have the highest reward. We analyze the bias and variance trade-offs of this new sampling gradient estimate, and show that the variance of our proposed approximation is lower than that of a naive MC approach.

For any given threshold $\alpha \in [0, 1]$, let $q_\alpha(\theta)$ denote the (upper) $\alpha-$quantile of the reward function $q_\alpha(\theta) \doteq \max_w \left\{ \sum_{z:r(z;x_{1:T},u_{1:T},\theta) \geq w} P(z) \leq \alpha \right\}$. This associates with the $\alpha-$quantile of the underlying intention with highest likelihood probability. We define an *intention-driven* probability distribution as follows:

$$Q_\alpha(z|u_{1:T}, x_{1:T}; \theta) = \frac{P(z)}{\alpha} \mathbf{1}\left\{ r(z; x_{1:T}, u_{1:T}, \theta) \geq q_\alpha(\theta) \right\}.$$

This quantity can be interpreted as a weighted distribution that only samples from the $\alpha\%$ of the most correct underlying intentions (i.e., the intentions that best explain the mentor demonstrations). The expected reward induced by the intention-driven distribution is given by:

$$\mathbb{E}_{z \sim Q_\alpha}[r(z; x_{1:T}, u_{1:T}, \theta)] = \mathbb{E}\left[ r(z; x_{1:T}, u_{1:T}, \theta) \,|\, r(z; x_{1:T}, u_{1:T}, \theta) \geq q_\alpha(\theta) \right], \quad (2)$$

which is equal to the conditional likelihood function of the $\alpha\%$ most correct intentions. In the financial risk literature, this metric is known as the *expected shortfall* (Rockafellar & Uryasev, 2000), and is typically used to evaluate the risky tail distribution of financial assets. While our setting is completely different, we will use tools developed for expected shortfall estimation in our approach.

By definition of $Q_\alpha$, one has the following inequality.

$$P(u_{1:T}|x_{1:T}; \theta) = \mathbb{E}[r(z; x_{1:T}, u_{1:T}, \theta)] \leq \mathbb{E}_{Q_\alpha}[r(z; x_{1:T}, u_{1:T}, \theta)] \leq \frac{1}{\alpha} P(u_{1:T}|x_{1:T}; \theta). \quad (3)$$

We propose to maximize $\mathbb{E}_{z \sim Q_\alpha}[r(z; x_{1:T}, u_{1:T}, \theta)]$ using Monte Carlo sampling techniques. Intuitively, since the support of $Q_\alpha$ is limited to the most likely $z$ values, estimating it using sampling has lower variance than estimating the original likelihood. We will further elaborate on this point technically later in the section. However, this comes at the cost of adding a bias $\sum_{z:Q_\alpha(z)=0} P(u_{1:T}|x_{1:T}, z; \theta) P(z)$. Empirically, we have found this procedure to work well.

To sample from $Q_\alpha$, we use empirical quantile estimation (Glynn, 1996). Let $z_1^{ord}, \ldots, z_N^{ord}$ denote the MC samples $z_1, \ldots, z_N$ sorted in descending order, according to the reward function $r(z; x_{1:T}, u_{1:T}, \theta)$. Let $N_\alpha = \lfloor \alpha N \rfloor$ be the number of samples corresponding to the $\alpha-$quantile. Then we have the following empirical estimate: $\mathbb{E}_{z \sim Q_\alpha}[r(z; x_{1:T}, u_{1:T}, \theta)] \approx \frac{1}{N_\alpha} \sum_{i=1}^{N_\alpha} r(z_i^{ord}; x_{1:T}, u_{1:T}, \theta)$. It has been shown in Theorem 1 of Glynn (1996) that under standard assumptions, the above expression is a consistent estimator of $\mathbb{E}_{z \sim Q_\alpha}[r(z; x_{1:T}, u_{1:T}, \theta)]$ with order $O(N^{-1/2})$, which we have shown above to be a lower bound to the likelihood function. For the special case of $N_\alpha = 1$ (when $\alpha = 1/N$), we can replace the sorting operation with a simple min operation, yielding a simple and intuitive algorithm – we choose the sampled $z$ with the lowest error for updating the parameters $\theta$.

In practice, maximizing the log-likelihood of the data is known to work well (Goodfellow et al., 2016). In our case, we correspondingly maximize $\mathbb{E}_{z \sim Q_\alpha}[\log r(z; x_{1:T}, u_{1:T}, \theta)]$, which, by the Jensen inequality[3], is a lower bound on $\log \mathbb{E}_{z \sim Q_\alpha}[r(z; x_{1:T}, u_{1:T}, \theta)]$. We therefore obtain the following gradient estimate $G_{N,\alpha}(\theta) := \frac{1}{N_\alpha} \sum_{i=1}^{N_\alpha} \nabla_\theta \log r(z_i^{ord}; x_{1:T}, u_{1:T}, \theta)$. We term this sampling technique as *intention-driven* sampling (IDS). Pseudocode is given in Algorithm 1.

**Optimistic Sampling:** To predict actions *at test time*, we first sample $z$ in the beginning of the episode and fix it, and then use the NN to predict $u_t = f(x_t, z)$ at every time step. While we could sample $z$ from $P(z)$, this might result in a $z$ value that has a low likelihood to reproduce the

---

[3]Since $\log$ is a monotonic function, all the order statistics of $r(z; x_{1:T}, u_{1:T}, \theta)$, including the quantile $q_\alpha(\theta)$, will not be affected.

demonstrations, i.e., a $z$ that has low reward. We observed this to be problematic in practice, and therefore devise an alternative approach, which we term *optimistic sampling*. We propose to store a set of the $K$ most recent $z$ values that obtained the highest reward during training, and at inference time sample a $z$ uniformly from this set. This corresponds to sampling $z$ from $Q_\alpha(z)$, averaged over the training data. Optimistic sampling dramatically improves the prediction performance in practice.

**Analysis:** By Theorem 4.2 of Hong & Liu (2009), $G_{N,\alpha}(\theta)$ is a consistent gradient estimator of the lower bound with asymptotic bias of $O(N^{-1/2})$. In Appendix 8, we deduce the following expression for the variance of $G_{N,\alpha}(\theta)$:

$$\mathbb{V}_{IDS} \approx \frac{1}{\alpha^2 N} \text{Var}(\nabla_\theta(\log r(z_1; x_{1:T}, u_{1:T}, \theta) - \overline{q}_\alpha(\theta))\mathbf{1}\{\log r(z_1; x_{1:T}, u_{1:T}, \theta) \geq \overline{q}_\alpha(\theta)\}).^4 \quad (4)$$

When $N_\alpha = N$, i.e., $\alpha = 1$, we obtain the variance of standard MC sampling , i.e., $\mathbb{V}_{MC} = \frac{1}{N}\text{Var}(\nabla_\theta \log r(z; x_{1:T}, u_{1:T}, \theta))$. On the other hand, when $\alpha \to 1/N$ the variance is bounded in $O(1/N^2)$. This result is due to the fact that **(i)** $|\nabla_\theta(\log r(z_1; x_{1:T}, u_{1:T}, \theta) - \overline{q}_\alpha(\theta))|\mathbf{1}\{\log r(z_1; x_{1:T}, u_{1:T}, \theta) \geq \overline{q}_\alpha(\theta)\}$ is bounded by a constant that is $O(\alpha)$,[5] **(ii)** $\text{Var}(\mathbf{1}\{\log r(z_1; x_{1:T}, u_{1:T}, \theta) \geq \overline{q}_\alpha(\theta)\}) = \alpha \cdot (1 - \alpha)$, and **(iii)** therefore $\mathbb{V}_{IDS} = O(\alpha \cdot (1 - \alpha)/(\alpha^2 N) \cdot \alpha^2) = O(N^{-2})$. Therefore, one can treat $\alpha$ as a nob to trade-off bias (see (3)) and variance (see (4)) in IDS.

---

**Algorithm 1:** IDS

---

**Input:** A minibatch of $K$ samples $\{u_t, x_t, \ldots, u_{t+K}, x_{t+K}\}$ from the same demonstration trajectory $\mathcal{T}^i$
**Output:** An update direction for $\theta$, and a sample from $Q_\alpha$

1 Sample $z_1, \ldots, z_N \sim P(z)$
2 Set $z^* = \arg\max_{z_i} P(u_{t:t+K}|x_{t:t+K}, z_i; \theta)$
3 **return** $\nabla_\theta \log P(u_{t:t+K}|x_{t:t+K}, z^*; \theta)$, and $z^*$

---

**Comparison to SNNs:** Broadly speaking, generalized EM algorithms (Tang & Salakhutdinov, 2013) can be seen as designing an importance sampling weight to reshape the sampling distribution, in order to lower the variance of the gradient estimate, by using entropy maximization or posterior distribution matching (more details can be found in Appendix 7). For the specific SNN algorithm of Tang & Salakhutdinov (2013) applied to our NN architecture, the importance weights correspond to a soft-max over the reward defined above. In IDS the importance weight is $w(z) = \frac{1}{\alpha}\mathbf{1}\{r(z; x_{1:T}, u_{1:T}, \theta) \geq q_\alpha(\theta)\}$, which for $N_\alpha = 1$ amounts to replacing the soft-max with a hard max. Interestingly, these two similar algorithms were developed from very different first principles. In practice, however, the IDS algorithm integrates naturally with optimistic sampling, which leads to significantly better performance, as we show in our experiments.

### 4.4 INTENTION DRIVEN SNN ARCHITECTURE FOR STOCHASTIC VISUAL ATTENTION

In this section we present *Intention-SNN* (henceforth I-SNN), an architecture that implements the stochastic intention as an attention over particular features in the image. This architecture is suitable for LfD domains where the visual observation contains information about multiple possible intentions, as in the object pick up task in Section 5.

Our I-SNN architecture is presented in Figure 1, and is comprised of three modules. The first module is a standard multi-layer (fully) convolutional neural network (CNN) feature extractor (Goodfellow et al., 2016), followed by a spatial softmax layer. The CNN maps the input image onto $C$ feature maps. The spatial softmax, introduced by Levine et al. (2015), calculates for each feature map in its input the corresponding $(x, y)$ position in the image where this feature is most active. Let $\phi_{c,i,j}$ denote the activation of feature map $c$ at coordinate $(i, j)$. The spatial softmax output for that feature is $(f_{c,x}, f_{c,y})$, where $f_{c,x} = \sum_{i,j} \exp(\phi_{c,i,j}) \cdot i / \sum_{i',j'} \exp(\phi_{c,i',j'})$, and $f_{c,y} = \sum_{i,j} \exp(\phi_{c,i,j}) \cdot j / \sum_{i',j'} \exp(\phi_{c,i',j'})$. Thus, the output of the spatial softmax is of dimensions $C \times 2$.

---

[4]Here $\overline{q}_\alpha(\theta))$ is the $\alpha$−quantile of the log reward function.

[5]Intuitively, when the threshold $\alpha$ approaches $1/N$, the quantile $\overline{q}_\alpha(\theta)$ increases (up to the maximum value of log reward) at rate $O(\alpha)$, leaving less and less room for log rewards above the quantile. Concretely, the random variable $|\log r(z_1; x_{1:T}, u_{1:T}, \theta) - \overline{q}_\alpha(\theta)|\mathbf{1}\{\log r(z_1; x_{1:T}, u_{1:T}, \theta) \geq \overline{q}_\alpha(\theta)\}$ converges to 0 at rate $O(\alpha)$. Consequently, this random variable is bounded by $O(\alpha)$. Analogous arguments can be applied for the case with gradients as well, more technical details can be found in Section 4 of Hong & Liu (2009).

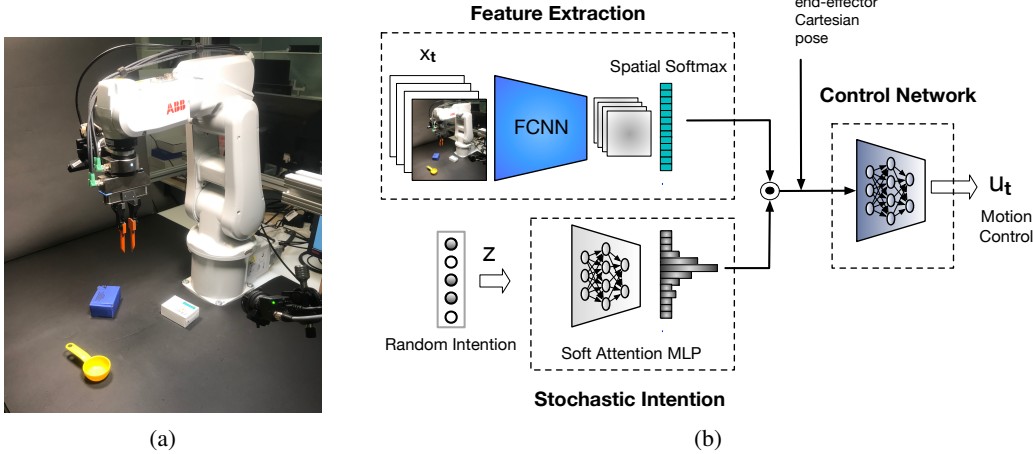

Figure 1: (a) 6-DOF IRB-120 robot and an example of a task configuration; (b) A schematic diagram of the I-SNN architecture.

The second module applies a stochastic soft attention over the spatial softmax features. We use a MLP to map the $M$-dimensional random intention vector $z$ onto a $C$-dimensional vector $w$. We then apply a softmax to obtain the attention weight vector $a \in \mathbb{R}^C$, $a_c = \exp(w_c)/\sum_{c'} \exp(w_{c'})$ (Xu et al., 2015). The attention weight is multiplied with the spatial softmax output to obtain the attention-modulated feature activations: $f'_{c,x} = f_{c,x} \cdot a_c$, and $f'_{c,y} = f_{c,y} \cdot a_c$, which are input to the control network along with the robot's $d$-dimensional vector of current pose. The control network is a standard MLP mapping $R^{C \times 2 + d}$ to $P(u)$.

The intuition behind this architecture is that the stochastic intention can 'select' which features are relevant for making a control decision, by giving them higher weights in the attention. When multiple objects are in the scene, each object would naturally be represented by different features, therefore the intention in this architecture can correspond to attending to a particular object[6].

## 5 EXPERIMENTS

We demonstrate the effectiveness of our approach on learning a reaching task with the IRB-120 robot (Figure 1b), where mentor trajectories are collected in the form of sequential image-action pairs. The main questions we seek to answer are: (1) Can our IDS learn to effectively reproduce multiple intentions in the demonstrations? (2) How does IDS approach compare to a standard deterministic NN approach for LfD? (3) How does training an I-SNN using IDS compare with training it using the SNN algorithm[7] of Tang & Salakhutdinov (2013)?

To maintain a fair comparison, we evaluated deterministic NN policies with identical structure as I-SNN except for the stochastic intention module (cf. Section 4.4, and Figure 1).

### 5.1 SNN IMPLEMENTATION DETAILS

In all our experiments we used the following parameters for the SNN training, which we found to work well consistently across different domains:(1) **Loss function**: we represent the output distribution as $\log P(u|h,x;\theta) \propto -\|f(x,h;\theta) - u\|_1$, where $f(x,h;\theta)$ is the output of the control network, as described in Section 4.4. This corresponds to an $L_1$ regression loss, which we found to perform better than the popular $L_2$ loss in our experiments. (2) **Monte Carlo samples**: we chose $N = 5$,

---

[6]We note that this architecture is not directly applicable for cases where two objects have exactly the same appearance, and therefore the same feature activation function. Such cases can be handled by adding spatial information to the features or the attention module, which will be investigated in future work.

[7]We also investigated using a conditional VAE (CVAE), however, despite extensive experimentation, we could not get the CVAE to work. We attribute this to the fact that the recognition network in a CVAE needs to map the image to a latent variable distribution that 'explains' the observed action. This mapping is complex, as it needs to understand from the image what goal the demonstration is aiming towards. Our approach, on the other hand, does not require a recognition module for reducing variance during training.

which we found to work well. Higher values resulted in degraded performance at test time, due to the higher bias (see Section 4). (3) **Intention** variable dimension: we chose $z \in \mathbb{R}^5$ for all experiments. We did not observe this parameter to be sensitive, and dimensions from 2 to 10 performed similarly. (4) $\mathbf{P}(\mathbf{z})$: a 5-dimensional vector of independent uniform multinomials in $\{0 : 4\}$.

## 5.2    REAL WORLD ROBOT REACHING TASK

In this task, depicted in Figure 1(a), the objective is to navigate the end-effector of robot to reach a point above one of 3 objects in the scene – a soap box, a blue electronic box and a measuring cup. We used a 6-DOF IRB-120 robot where the control is a 3-dimensional Cartesian vector applied to the the end effector $u_t = (d\vec{x_t}, d\vec{y_t}, d\vec{z_t})$. The observations consists of (1) a $480 \times 640$ RGB image of the scene (further cropped and resized to $64 \times 64$ resolution) using a point-grey camera mounted at a fixed angle; (2) the 3 dimensional end-effector Cartesian pose (see Figure 1 for more details).

**Data Collection:**    We denote a specific scene arrangement of object placements and initial end-effector pose as a *task configuration*. Task configurations were randomly generated by arbitrarily placing the three objects on the work bench and randomly initializing the end-effector pose. Once a task configuration is generated, the position of the objects remain fixed for the entire episode. For each task configuration we collect 3 demonstrations, each generated by a human navigating the end-effector using a 3DConnexion space-mouse to reach one of the objects. At each time step $t$, the observation $o_t$ together with the Cartesian control signal $u_t$ (see above) are recorded. We collected demonstrations from 468 different task configurations, for a total of 1404 demonstration trajectories.

**Training:**    We compare using IDS and the SNN algorithm of Tang & Salakhutdinov (2013) (henceforth SNN) for training the I-SNN architecture. To reduce the training time, we pre-trained the weights of the convolutional layers in the Feature Extraction module reusing the weights learned in the deterministic NN model. For optimization, we also used Adam (Kingma & Ba, 2014), with the default parameters using 90% of the data set for training and 10% for validation. .

**Evaluation:**    We evaluate each model on 10 randomly generated task configurations, with 20 trials on each task configuration, for a total of 200 trials. We run the model until it either succeeds to reach an object or fails. For the IDS algorithm, we used optimistic sampling. For SNN, we experimented with both optimistic and uniform sampling, however the latter resulted in a better overall performance.

Table 1 shows the overall success rate for every model across all 200 trials. The deterministic NN model succeeded in reaching one of the objects only in 3 task configurations (and kept on reaching that same object for the 20 evaluations in each), and failed on the other 7 task configurations due to the averaging problem. The stochastic algorithms performed significantly better by learning multiple modes of the problem. IDS significantly out-performed the SNN algorithm, as we explain below.

We evaluate the mode learning ability by counting, for each task configuration, how many *different* objects were reached, as depicted in Table 1 (last four columns). As expected, the deterministic NN could not reach more than one object. I-SNN trained by SNN algorithm, on the other hand, could reach two different (but fixed) objects for 6 task configurations, and all three objects for the rest. The best performance is achieved by I-SNN trained by IDS, reaching all three objects in all the tasks, thereby demonstrating a strong ability to learn all the modes in the data. In the appendix (Figure 3), we show a histogram of reaching the different objects that demonstrates that IDS learned a near-uniform distribution over the modes. Additionally, in the appendix (Figure 5) we visualize the features that I-SNN attends to, showing that in each episode the model consistently attends to the same object throughout the execution.

In Figure 2 we explain the superior performance of IDS over SNN. Since IDS focuses only on the best samples (through the $\max$) compared to SNN which gives weight also to non best samples (through the soft-$\max$), IDS better 'tunes' the network for the best samples. Combined with optimistic sampling, which draws these best samples during execution, this leads to better results.

## 6    CONCLUSION

We presented an approach for learning from demonstrations that contain multiple modes of performing the same task. Our method is based on stochastic neural networks, and represents the mode

Table 1: The first column shows the overall success rate for every model calculated over 10 different task configurations, and 20 trials each, for a total of 200 trials. The last four columns show the un-normalized distribution of modes learned across 10 task configurations.

| | Overall Success Rate | 1 object | 2 objects | 3 objects | Fail |
|---|---|---|---|---|---|
| **deterministic NN** | 30% | 3 | 0 | 0 | 7 |
| **SNN** | 60.5% | 0 | 6 | 4 | 0 |
| **IDS** | **98.5%** | 0 | 0 | **10** | 0 |

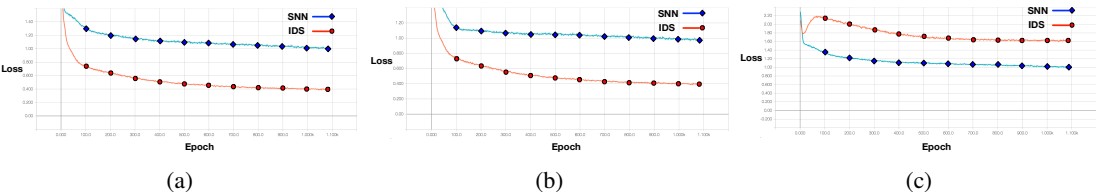

| (a) | (b) | (c) |

Figure 2: Comparison of IDS and SNN algorithms. We plot three different errors during training (on the training data), for the same model trained using IDS and SNN algorithm. Left: the respective training loss for each method. Since the max in IDS upper bounds the softmax in SNN, the loss plot for IDS lower bounds SNN. Middle: the IDS loss on the training data, for both models. Since the SNN is trained on a different loss function (softmax), its performance is worse. This shows an important point: if, at test time, we use optimistic sampling to sample $z$ from best samples during training, we should expect IDS to perform better than SNN. Right: the average log-likelihood loss during training. The SNN wins here, since the softmax encourages to increase the likelihood of 'incorrect' $z$ values. This provides additional motivation for using optimistic sampling.

of performing the task by a stochastic vector – the *intention*, which is given as input to a feedforward neural network. We presented a simple and efficient algorithm for training our models, and a particular implementation suitable for vision-based inputs. As we demonstrated in real-robot experiments, our method can reliably learn to reproduce the different modes in the demonstration data, and outperforms standard approaches in cases where such different modes exist.

In future work we intend to investigate the extension of this approach to more complex manipulation tasks such as grasping and assembly, and domains with a very large number of objects in the scene. An interesting point in our model is tying the features to the intention by an attention mechanism, and we intend to further investigate recurrent attention mechanisms (Xu et al., 2015) that could offer better generalization at inference time.

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

# 7 GENERALIZED EM ALGORITHM FOR LEARNING SNNS VIA IMPORTANCE SAMPLING

We review the work of Tang & Salakhutdinov (2013) as relevant for our setting. For the importance sampling based approach, instead of maximizing the log likelihood function, consider maximizing the following evidence lower bound (ELBO):

$$
\begin{aligned}
F(Q,\theta) &= E_Q \left[ \log \frac{P(u_{1:T}, z|x_{1:T}; \theta)}{Q(z|x_{1:T}, u_{1:T})} \right] \\
&\leq E_{P(\cdot|x_{1:T}, u_{1:T}; \theta)} \left[ \log P(y_{1:T}|x_{1:T}, \theta) \right] \\
&\leq \log P(y_{1:T}|x_{1:T}, \theta),
\end{aligned}
$$

where $F$ is the Kullback Liebler divergence between $P(u_{1:T}|z, x_{1:T}; \theta)$ and $Q(z|u_{1:T}, x_{1:T})$ given as follows:

$$
F(Q,\theta) = -D_{KL}(Q||P(\cdot|x_{1:T}, u_{1:T}; \theta)) + \log P(y_{1:T}|x_{1:T}, \theta).
$$

Most importantly, it has also been shown in Theorem 2 of Neal & Hinton (1998) that if $Q$ and $\theta$ form a pair of local maximizer to $F$, then $\theta$ is also a local maximum of the original likelihood maximization problem. To maximize $F$ w.r.t $Q$, one has the closed form solution based on Bayes theorem:

$$
\begin{aligned}
Q^*(z|u_{1:T}, x_{1:T}; \theta_{\text{old}}) &= P(z|y_{1:T}, x_{1:T}, \theta) \\
&= \frac{P(y_{1:T}|z, x_{1:T}, \theta_{\text{old}})P(z)}{P(y_{1:T}|x_{1:T}, \theta_{\text{old}})} \\
&\approx \frac{P(y_{1:T}|z, x_{1:T}, \theta_{\text{old}})P(z)}{\frac{1}{N}\sum_{i=1}^{N} P(y_{1:T}|z_j, x_{1:T}, \theta_{\text{old}})}.
\end{aligned}
$$

Here, $\{z_1, \ldots, z_N\}$ is a sequence of latent random variables sampled i.i.d. from the distribution $P(z)$.

Given parameter $\theta$, denoted by $\theta_{old}$, immediately the posterior distribution $Q$ that maximizes $F$ is given by: $Q^*(z|x_{1:T}, u_{1:T}) = P(z|x_{1:T}, u_{1:T}; \theta_{\text{old}})$. In this case, the above loss function is equivalent to the complete data log-likelihood

$$
\ell^*(\theta, \theta_{\text{old}}) := \mathbb{E}_{P(\cdot|u_{1:T}, x_{1:T}; \theta_{\text{old}})} \left[ \log \frac{P(x_{1:T}, z|u_{1:T}; \theta)}{P(z|x_{1:T}, u_{1:T}; \theta_{\text{old}})} \right],
$$

which is a lower bound of the log likelihood. Furthermore, if $\theta = \theta_{\text{old}}$, then clearly $\ell^*(\theta_{\text{old}}, \theta_{\text{old}})$ is equal to the log-likelihood $\log P(y_{1:T}|x_{1:T}, \theta_{\text{old}})$.

Tang & Salakhutdinov (2013) present a generalized EM algorithm to train a SNN. In the E-step, the following approximate posterior distribution is used:

$$
\hat{Q}(z|u_{1:T}, x_{1:T}; \theta_{\text{old}}) := \bar{r}(z; x_{1:T}, y_{1:T}, \theta_{\text{old}})P(z),
$$

where

$$
\bar{r}(z; x_{1:T}, y_{1:T}, \theta_{\text{old}}) = \frac{r(z; x_{1:T}, u_{1:T}, \theta_{\text{old}})}{\frac{1}{N}\sum_{i=1}^{N} r(z_i; x_{1:T}, u_{1:T}, \theta_{\text{old}})}
$$

is the the importance sampling weight. Recall that for our distribution model, $r(z; x_{1:T}, u_{1:T}, \theta_{\text{old}}) \propto \exp(-d(f(x, z; \theta), u))$, therefore we obtain that the importance weights correspond to a soft-max over the prediction error.

In the M-step, the $\theta$ parameters are updated with the gradient vector with respect to the following optimization: $\theta \in \arg\max_{\theta \in \Theta} \hat{\ell}(\theta, \theta_{\text{old}})$, where

$$
\hat{\ell}(\theta, \theta_{\text{old}}) = \frac{1}{N}\sum_{i=1}^{N} \bar{r}(z_i; x_{1:T}, y_{1:T}, \theta_{\text{old}}) \log P(y_{1:T}, z_i|x_{1:T}, \theta)
$$

is the empirical expected log likelihood, and $\hat{Q}$ is the posterior distribution from the E-step. Here we drop the last term in $F$ because in our case $Q$ that does not depend on $\theta$. Correspondingly, the gradient estimate is given by:

$$
\nabla_\theta \hat{\ell}(\theta, \theta_{\text{old}}) = \frac{1}{N}\sum_{i=1}^{N} \bar{r}(z_i) \nabla_\theta \log r(z_i; x_{1:T}, u_{1:T}, \theta),
$$

the equality is due to the facts that

$$\log P(y_{1:T}, z | x_{1:T}, \theta) = \log r(z; x_{1:T}, y_{1:T}, \theta) + \log P(z)$$

and distribution $P(z)$ is independent of $\theta$.

To better understand this estimator, we will analyze the bias and variance of the gradient estimator. Based on the construction of importance sampling weight, immediately the gradient estimator is consistent. Furthermore, under certain regular assumptions, the bias is $O(N^{-1/2})$. (This means the gradient estimator is asymptotically unbiased.) Furthermore, the variance of this estimator is given by

$$\mathbb{V}_{IS}(\theta, \theta_{\text{old}}) = \frac{1}{L} \left( \int_z v(z; \theta) dP(z) - (\nabla_\theta \ell^*(\theta, \theta_{\text{old}}))^2 \right),$$

where the integrand is given by $v(z; \theta) = \bar{r}(z; x_{1:T}, y_{1:T}, \theta_{\text{old}}) \cdot (\nabla_\theta \log r(z; x_{1:T}, u_{1:T}, \theta))^2 \geq 0$.

## 8 TECHNICAL PROOF OF THE VARIANCE IN INTENTION-DRIVEN SAMPLING

In this section, we study the variance of the gradient estimate of $\text{CVaR}(R(z; \theta))$. This proof follows analogously from the analysis in Section 4.3 of Hong & Liu (2009) for the case of estimating asymptotic variance in gradient. Here we use $R(z; \theta)$ as the shorthand notation of the log reward function $\log r(z; x_{1:T}, y_{1:T}, \theta)$. For any given cut-off level $\alpha$ and sample size $n$, consider the following update formula of the CVaR gradient estimate

$$G_N = \frac{1}{\alpha N} \sum_{i=1}^N \nabla_\theta R(z_i; \theta) S(z_i),$$

where $S(z_i) = \mathbf{1}\{R(z_i; \theta) \geq \bar{q}_\alpha(\theta)\}$, $\forall i$, and $\{z_1, \dots, z_N\}$ is sampled in an i.i.d. fashion from $P(h)$. Notice that

$$\begin{aligned}
\alpha^2 \text{Var}(G_N) =& \frac{1}{N} \text{Var}(\nabla_\theta R(z_1; \theta) S(z_1)) + \left( 1 - \frac{1}{N} \right) \cdot \text{Cov}(\nabla_\theta R(z_1; \theta) S(z_1), \nabla_\theta R(z_2; \theta) S(z_2)) \\
=& \frac{1}{N} \text{Var}(\nabla_\theta R(z_1; \theta) S(z_1)) + \left( 1 - \frac{1}{N} \right) \cdot \left[ \mathbb{E}[\nabla_\theta R(z_1; \theta) \nabla_\theta R(z_2; \theta) S(z_1) S(z_2)] \right. \\
& \left. - \mathbb{E}[\nabla_\theta R(z_1; \theta) S(z_1)]^2 \right].
\end{aligned}$$

The correlation comes from the fact that quantile is defined based on the order statistics of the reward, see Section 4 of Hong & Liu (2009) for more details. Now notice the following property:

$$\begin{aligned}
S(z_1) \cdot S(z_2) =& \mathbf{1}\{R(z_1; \theta) \geq \bar{q}_\alpha(\theta), R(z_2; \theta) \geq \bar{q}_\alpha(\theta)\} \\
=& \mathbf{1}\{R(z_1; \theta) > L_{\lceil N\alpha \rceil - 1:N}, R(z_2; \theta) > L_{\lceil N\alpha \rceil - 1:N}\} \\
=& \mathbf{1}\{R(z_1; \theta) > L_{\lceil N\alpha \rceil - 1:N-2}\} \mathbf{1}\{R(z_2; \theta) > L_{\lceil N\alpha \rceil - 1:N-2}\}
\end{aligned}$$

where $L_{i:N}$ is the $i$-th order statistic from the $N$ observations of reward $\{R(z_i; \theta)\}_{i=1}^N$. Under the event that $R(z_1; \theta) > L_{\lceil N\alpha \rceil - 1:N-2}$ and $R(z_2; \theta) > L_{\lceil N\alpha \rceil - 1:N-2}$, by the definition of the order statistic and by the i.i.d. assumption of $\{z_1, \dots, z_N\}$, one can deduce that $L_{\lceil N\alpha \rceil - 1:N-2}$ is independent of $R(z_1; \theta)$ and $R(z_2; \theta)$. Equipped with this condition, one has the following expression:

$$\begin{aligned}
\mathbb{E}[\nabla_\theta R(z_1; \theta) \nabla_\theta R(z_2; \theta) S(z_1) S(z_2)] =& \mathbb{E}[f^2(L_{\lceil N\alpha \rceil - 1:N-2})] \\
=& \text{Var}(f(L_{\lceil N\alpha \rceil - 1:N-2})) + \mathbb{E}[f(L_{\lceil N\alpha \rceil - 1:N-2})]^2,
\end{aligned}$$

where $f(L_{\lceil N\alpha \rceil - 1:N-2}) := \mathbb{E}[\nabla_\theta R(z_1; \theta) S(z_1) \mid L_{\lceil N\alpha \rceil - 1:N-2}]$ is the conditional expectation of $\nabla_\theta R(z; \theta) S(z)$ w.r.t. random variable $L_{\lceil N\alpha \rceil - 1:N-2}$. The first equality follows from the fact that $z_1$, and $z_2$ are i.i.d. random variables that are independent of $L_{\lceil N\alpha \rceil - 1:N-2}$.

On the other hand, following the same lines of analysis, one can also show that

$$\mathbb{E}[\nabla_\theta R(z_1; \theta) S(z_1)]) = \mathbb{E}[f(L_{\lceil N\alpha \rceil - 1:N-1})].$$

Therefore, combining all the above analysis together, one has the following expression:

$$\alpha^2 \text{Var}(G_N) = \frac{1}{N}\text{Var}(\nabla_\theta R(z_1;\theta)S(z_1) + \left(1 - \frac{1}{N}\right) \cdot \left[\text{Var}(f(L_{\lceil N\alpha\rceil-1:N-2})) + \mathbb{E}[f(L_{\lceil N\alpha\rceil-1:N-2})]^2]\right.$$
$$\left. - \mathbb{E}[f(L_{\lceil N\alpha\rceil-1:N-1})]^2\right].$$

From the results in Proposition 4.3 to 4.4 of Hong & Liu (2009), with $\overline{q}'_\alpha(\theta) = \nabla_\theta \overline{q}_\alpha(\theta)$ denote the gradient of the quantile, we have that

$$\text{Var}(f(L_{\lceil N\alpha\rceil-1:N-2})) = \frac{\alpha(1-\alpha)(\overline{q}'_\alpha(\theta))^2}{N-1} + O(N^{-1/2}),$$

and

$$\mathbb{E}[f(L_{\lceil N\alpha\rceil-1:N-2})]^2 - \mathbb{E}[f(L_{\lceil N\alpha\rceil-1:N-1})]^2$$
$$= O(N^{-1/2}) - \frac{2(1-\alpha)}{N-1}\overline{q}'_\alpha(\theta)\mathbb{E}[\nabla_\theta R(z_1;\theta)\mathbf{1}\{R(z_1;\theta) \geq \overline{q}'_\alpha(\theta)\}].$$

Therefore, the variance of $G_N$ can be expressed as:

$$\alpha^2 \text{Var}(G_N) = \frac{1}{N}\text{Var}(\nabla_\theta R(z_1;\theta)S(z_1)) + \left(1 - \frac{1}{N}\right) \cdot \left[\frac{\alpha(1-\alpha)(\overline{q}'_\alpha(\theta))^2}{N-1}\right.$$
$$\left. - \frac{2(1-\alpha)}{N-1}\overline{q}'_\alpha(\theta)\mathbb{E}[\nabla_\theta R(z_1;\theta)S(z_1)]\right] + O(N^{-1/2})$$
$$= \frac{1}{N}\left[\mathbb{E}[(\nabla_\theta R(z_1;\theta))^2 S(z_1)] + \alpha(\overline{q}'_\alpha(\theta))^2\right.$$
$$\left. - 2\overline{q}'_\alpha(\theta)\mathbb{E}[(\nabla_\theta R(z_1;\theta))S(z_1)]\right] - \left[\mathbb{E}[(\nabla_\theta R(z_1;\theta))^2 S(z_1)]\right.$$
$$\left. + \alpha^2(\overline{q}'_\alpha(\theta))^2 - 2\alpha q'_\alpha(\theta)\mathbb{E}[(\nabla_\theta R(z_1;\theta))S(z_1)]\right] + O(N^{-1/2})$$
$$= \frac{1}{N}\left[\mathbb{E}[(\nabla_\theta R(z_1;\theta) - \overline{q}'_\alpha(\theta))^2 S(z_1)] - \mathbb{E}[(\nabla_\theta R(z_1;\theta) - \overline{q}'_\alpha(\theta))S(z_1)]^2\right] + O(N^{-1/2})$$
$$= \frac{1}{N}\text{Var}([\nabla_\theta R(z_1;\theta) - \overline{q}'_\alpha(\theta)]\mathbf{1}\{R(z_1;\theta) \geq \overline{q}_\alpha(\theta)\}) + O(N^{-1/2}).$$

This provides the variance of intention-driven sampling gradient estimate.

## 9 ADDITIONAL MATERIAL

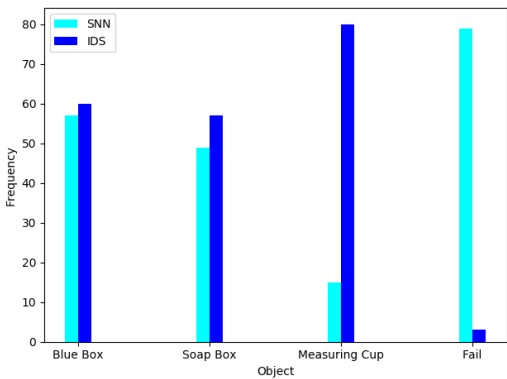

Figure 3: Raw histogram of the learned modes and failure cases for I-SNN architecture trained by SNN v.s. IDS algorithms.

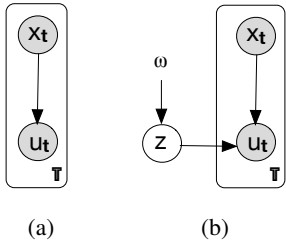

(a)                    (b)

Figure 4: (a) Most LfD approaches often learn multimodal representations as a function of current state (or history of states captured by a recurrent neural network). In such models, task level intention is not guaranteed to be consistently inferred at every step throughout the task execution; (b) in contrast I-SNN samples and uniformly commits to the same mode at the task level throughout the task execution.

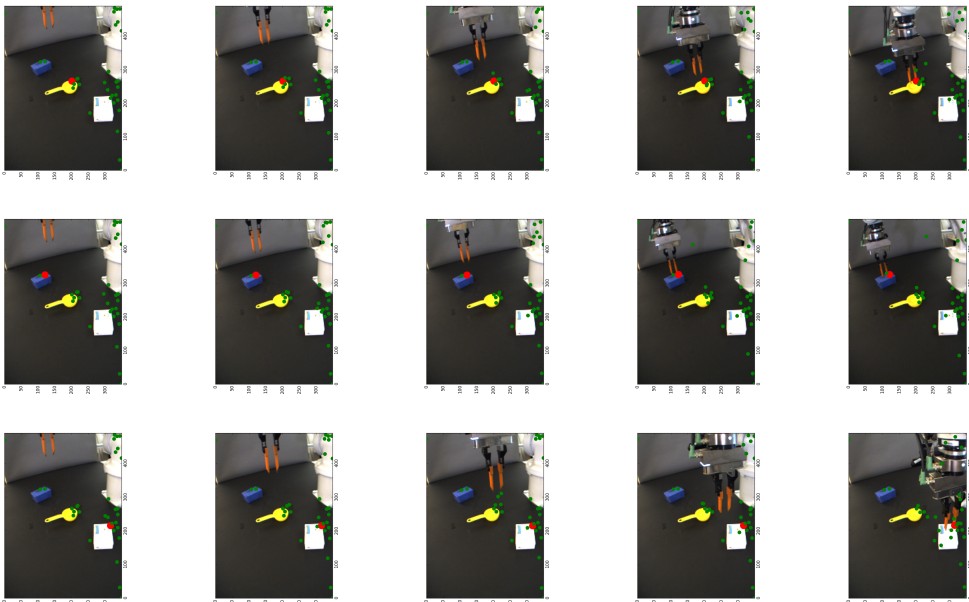

Figure 5: Visualization of the stochastic Intention network: every row shows 5 snapshots of a trajectory generated by running the I-SNN trained by the IDS algorithm. Each run was generating by randomly sampling an intention at the beginning and using it throughout the run. Smaller green circles show the 32 coordinates outputted by the spatial softmax layer. The larger red circle shows the top spatial softmax feature that received the highest weight from the soft attention generated by the Stochastic Intention Network. Note that for each run, the model consistently attends to the same mode that it randomly selected at the beginning of the run.

## 10  ADDITIONAL EXPERIMENTS

In this section we present simulation results that compare our IDS approach with a state-of-the-art CVAE based approach (Sohn et al., 2015).

We consider a simplified task which we term 'Predict the Goal', depicted in Figure 6. Given an image with N randomly positioned targets with different colors, the task is to predict the location (i.e., x-y position) of one of them. For training, we randomly selected one of the targets and provide its location as the supervisory signal. This task captures the essence of the robotic task in the paper – image input and a low dimensional multi-modal output (with N modes). It simplifies the image processing, and the fact that there is no trajectory – this is a single step decision making problem.

For our IDS algorithm, we used the I-SNN architecture as described in Figure 1, with a single conv layer (and without the additional robot pose input). The output is the 2-dimensional target position.

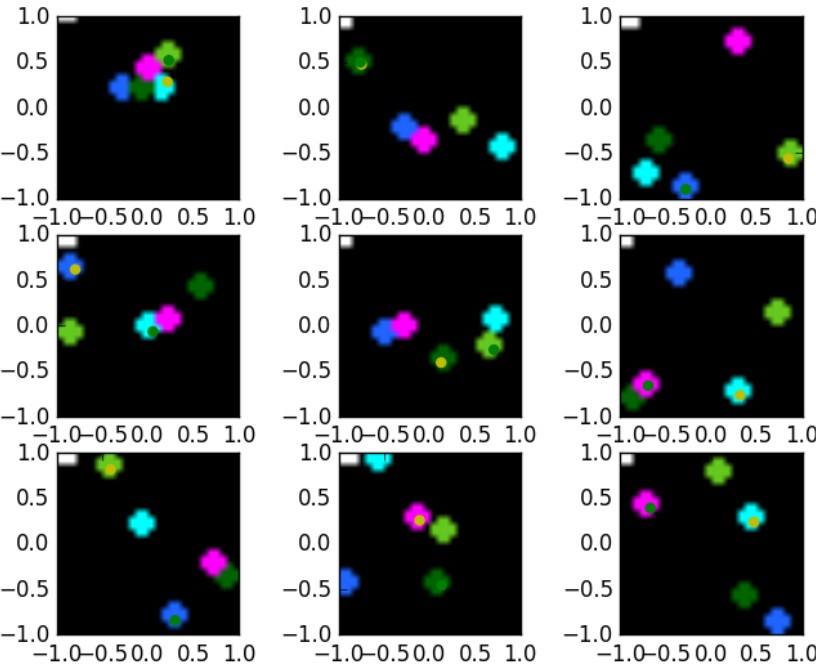

Figure 6: Predict the Goal Domain: in an image with $N$ randomly positioned, different colored targets, the task is to predict the center of one of the targets. The figure shows 9 random instances of a domain with $N = 5$ targets. We also plot the training target positions (dark green dots, selected uniformly among the targets), and the predictions of the trained I-SNN (yellow dots). Note that the predictions do not have to match the training targets, but have to be centered on some target in the image.

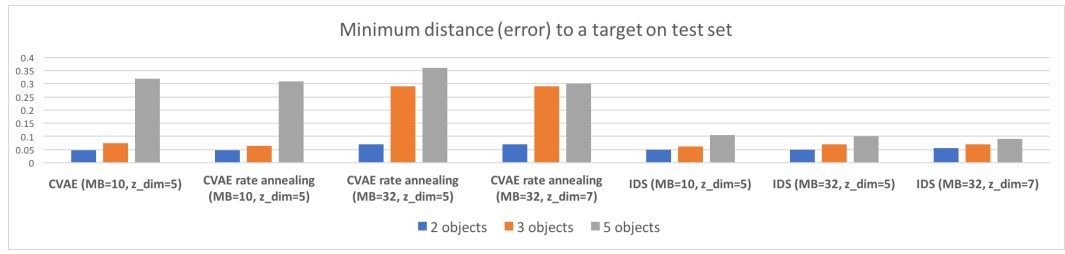

Figure 7: Results for Predict the Goal. We compare the CVAE with and without rate annealing, with different minibatch sizes, and with different sizes of the latent vector $z$ to IDS with the same parameters.

For the CVAE, the generation network $P(u|x, z)$ is the same I-SNN. For the recognition network $q(z|x, u)$ we used an MLP mapping the spatial-softmax output and the target position to the mean and std of $z$. For the conditional prior network $p(z|x)$ we used an MLP mapping the spatial-softmax output to the mean and std of $z$. Following the work of Sohn et al. (2015), we added to the training loss a term $\text{KL}(q(z|x, u)\|p(z|x))$.

To make the comparison fair, we chose the latent variable $z$ in IDS to be a standard Gaussian, the same as for the CVAE. All network sizes and training parameters were the same for both methods, and we did not apply any pretraining of the conv layer.

For evaluating performance, we measure the shortest distance from the prediction to one of the target positions, on a held-out test set. This error should go to zero if the model predicts one of the targets accurately.

Our results are reported in Figure 7. We have tried various CVAE parameter settings (such as the minibatch sizes and dimension of $z$ reported here, which the CVAE was sensitive to, among other parameters such MLP architectures and learning rates), and also tried annealing the KL term in the cost. The CVAE works well for $N = 2$ targets, and with careful tuning also for $N = 3$, but we could not get it to work for $N = 5$ targets. The IDS approach, on the other hand, worked well and robustly for all values of $N$ we tried. The convergence of IDS in all cases was also an order of magnitude faster than the CVAE.

