# OpenReview forum: "Imitation Learning from Visual Data with Multiple Intentions"
_ICLR.cc/2018/Conference — Accept (Poster)_

### Official Review · AnonReviewer2 · 2017-11-21
**Latent variable model for multi-modal imitation learning**

**Rating:** 6
**Confidence:** 4

**Review:**

The authors propose a new sampling based approach for inference in latent variable models. They apply this approach to multi-modal (several "intentions") imitation learning and demonstrate for a real visual robotics task that the proposed framework works better than deterministic neural networks and stochastic neural networks.

The proposed objective is based upon sampling from the latent prior and truncating to the largest alpha-percentile likelihood values sampled. The scheme is motivated by the fact that this estimator has a lower variance than pure sampling from the prior. The objective to be maximized is a lower bound to 1/alpha * the likelihood.

Quality: The empirical results (including a video of an actual robotic arm system performing the task) looks good. This reviewer is a bit sceptical to the methodology. I am not convinced that the proposed bound will have low enough variance. It is mentioned in a footnote that variational autoencoders were tested but that they failed. Since the variational bound has much better sampling properties (due to recognition network, reparameterization trick and bounding to get log likelihoods instead of likelihoods) it is hard to believe that it is harder to get to work than the proposed framework. Also, the recently proposed continuous relaxation of random variables seemed relevant.

Clarity: The paper is fairly clearly written but there are many steps of engineering that somewhat dilutes the methodological contribution.

Significance: Hard to say. New method proposed and shown to work well in one case. Too early to tell about significance.

Pro:
1. Challenging and relevant problem solved better than other approaches.
2. New latent variable model bound that might work better than classic approaches.
Con:
1. Not entirely convincing that it should work better than already existing methods.
2. Missing some investigation of the properties of the estimator on simple problem to be compared to standard methods.

---

### Official Review · AnonReviewer1 · 2017-11-26
**This work shows how to learn several modalities using Imitation learning from visual data using stochastic Neural Network.**

**Rating:** 4
**Confidence:** 3

**Review:**

The authors provide a method for learning from demonstrations where several modalities of the same task are given. The authors argue that in the case where several demonstrations exists and a deterministic (i.e., regular network) is given, the network learns some average policy from the demonstrations.

The paper begins with the authors stating the motivation and problem of how to program robots to do a task based only on demonstrations rather on explicit modeling or programming. They put the this specific work in the right context of imitation learning and IRL. Afterward, the authors argue that deterministic network cannot adequately several modalities. The authors cover in Section 2 related topics, and indeed the relevant literature includes behavioral cloning, IRL , Imitation learning, GAIL, and VAEs. I find that recent paper by Tamar et al 2016. on Value Iteration Networks is highly relevant to this work: the authors there learn similar tasks (i.e., similar modalities) using the same network. Even the control task is very similar to the current proposed task in this paper.

The authors argue that their contribution is 3-fold: (1) does not require robot  rollouts, (2) does not require label for a task, (3) work within raw image inputs. Again, Tamar et al. 2016 deals with this 3 points.

I went over the math. It seems right and valid. Indeed, SNN is a good choice for adding (Bayesian) context to a task. Also, I see the advantage of referring only to the "good" quantiles when needed. It is indeed a good method for dealing with the variance.

I must say that I was impressed with the authors making the robot succeed in the tasks in hand (although reaching to an object is fairly simple task).

My concerns are as follows:
1) Seems like that the given trajectories are naturally divided with different tasks, i.e., a single trajectory consists only a single task. For me, this is not the pain point in this tasks. the pain point is knowing when tasks are begin and end.
2) I'm not sure, and I haven't seen evidence in the paper (or other references) that SNN is the only (optimal?) method for this context. Why not adding (non Bayesian) context (not label) to the task will not work as well?
3) the robot task is impressive. but proving the point, and for the ease of comparing to different tasks, and since we want to show the validity of the work on more than 200 trials, isn't showing the task on some simulation is better for understanding the different regimes that this method has advantage? I know how hard is to make robotic tasks work...
4) I’m not sure that the comparison of the suggested architecture to one without any underlying additional variable Z or context (i.e., non-Bayesian setup) is fair. "Vanilla" NN indeed may fail miserably . So, the comparison should be to any other work that can deal with "similar environment but different details".

To summarize, I like the work and I can see clearly the motivation. But I think some more work is needed in this work: comparing to the right current state of the art, and show that in principal (by demonstrating on other simpler simulations domains) that this method is better than other methods.

---

### Official Review · AnonReviewer3 · 2017-11-28

**Rating:** 6
**Confidence:** 4

**Review:**

This paper focuses on imitation learning with intentions sampled
from a multi-modal distribution. The papers encode the mode as a hidden
variable in a stochastic neural network and suggest stepping around posterior
inference over this hidden variable (which is generally required to
do efficient maximum likelihood) with a biased importance
sampling estimator. Lastly, they incorporate attention for large visual inputs.

The unimodal claim for distribution without randomness is weak. The distribution
could be replaced with a normalizing flow. The use of a latent variable
in this setting makes intuitive sense, but I don't think multimodality motivates it.

Moreover, it really felt like the biased importance sampling approach should be
compared to a formal inference scheme. I can see how it adds value over sampling
from the prior, but it's unclear if it has value over a modern approximate inference
scheme like a black box variational inference algorithm or stochastic gradient MCMC.

How important is using the pretrained weights from the deterministic RNN?

Finally, I'd also be curious about how much added value you get from having
access to extra rollouts.

---

### Author Response · Authors · 2017-12-19
**Review response**

We thank the reviewers for the thoughtful comments.

The paper has been updated with additional simulation experiments.

We start by describing the additional experiments, and then address each reviewer separately.

Following the reviewers suggestions, we include results that compare our approach to a state-of-the-art conditional VAE on a simulated domain. These results were omitted in our initial submission with the interest of keeping the paper at the suggested page limits. We briefly summarize the results here, see Appendix D for more details.

The experiments were conducted on a simple simulated domain: given an image with N randomly positioned targets with different colors, predict the location (i.e., x-y position) of one of them. For training, we randomly selected one of the targets and provided its location as the supervisory signal.
This task captures the essence of the robotic task in the paper - image input and a low dimensional multi-modal output (with N modes). It simplifies the image processing, and the fact that there is no trajectory - it’s a single step decision making problem.

To make the comparison fair, we chose the latent variable z in IDS to be a standard Gaussian, just as for the CVAE. All network sizes and training parameters were the same for both methods (except for the additional recognition and conditional prior network for CVAE), and we did not apply any pretraining to the conv layer.

We have tried various CVAE parameter settings, and also annealing of the KL term in the cost. The CVAE works well for N=2 targets, and with careful tuning also for N=3, but despite genuine efforts we could not get it to work for N=5 targets. These results actually motivated us to follow the IDS approach in the first place, which worked well and robustly for all values of N we tried. The convergence of IDS in all cases was also an order of magnitude faster.

These results show that:
1) In some domains our IDS algorithm works significantly better than state of the art algorithms for variational inference.
2) Pretraining is not required for our approach (though it definitely helps speed it up).

While it could definitely be the case that with more parameter tuning, or that by applying other improvements to CVAEs such as normalizing flows we could make them work in this task, we believe that the simplicity of our approach and its robust performance is worth reporting.

A similar result was recently reported by Fragkiadaki et al. (2017), comparing CVAEs to backpropping through top-k samples in video prediction. Our contribution, compared to that work, is grounding this method in a formal mathematical treatment, proposing optimistic sampling which significantly improves its performance, and showing its importance in a real world robotic imitation learning domain.

References:
Fragkiadaki, Katerina, et al. "Motion Prediction Under Multimodality with Conditional Stochastic Networks." arXiv preprint arXiv:1705.02082 (2017).



AnonReviewer1:

Comparison to value iteration networks (VIN):
The VIN work does not consider multiple modes in the data, which is the main focus in our work. In particular, the target position in the VIN paper is *explicitly provided* as a separate image channel of the input, and the VIN output is deterministic - it cannot reproduce multiple modes of reaching to different targets. Thus, VINs cannot solve the problems we tackle in this paper.

Extending VINs with latent variables or using VINs inside a generative model is an interesting direction, but one that would require a separate investigation.

We believe our related work section covers most relevant works on imitation learning with multiple intentions/modes in the data.

Answers to specific comments:
1) In our setting (and in many realistic industrial setting) knowing when the demonstrations start and end is trivial, as the demonstrator records demonstrations sequentially.
2) Adding context would require to either label the context or infer it. Labelling adds burden on the demonstrator, which we wish to minimize. Inferring the context is the approach we pursue, and we added additional experiments comparing our approach to a state of the art variational inference method.
3+4) See above for additional simulation results.

AnonReviewer2:
See above - we added a comparison with conditional VAEs.

AnonReviewer3:

Pretrained weights - see above. Pretraining is not necessary, but definitely helps speed up training.

Extra rollouts: We did not fully understand this comment. Generally rollouts are better understood in the context of an RL setting, however our approach is not RL and thus no rollout is involved. While extra RL rollouts can be used to improve the policy, in many realistic scenarios taking extra rollouts on the robot can be costly/unsafe/time consuming.

---

### Decision · Program_Chairs · 2018-01-29
**ICLR 2018 Conference Acceptance Decision**

**Decision:**

Accept (Poster)

**Comment:**

This paper presents a sampling inference method for learning in multi-modal demonstration scenarios. Reference to imitation learning causes some confusion with the IRL domain, where this terminology is usually encountered. Providing a real application to robot reaching, while a relatively simple task in robotics, increases the difficulty and complexity of the demonstration. That makes it impressive, but also difficult to unpick the contributions and reproduce even the first demonstration. It's understandable at a meeting on learning representations that the reviewers wanted to understand why existing methods for learning multi-modal distributions would not work, and get a better understanding of the tradeoffs and limitations of the proposed method. The CVAE comparison added to the appendix during the rebuttal period just pushed this paper over the bar. The demonstration is simplified, so much easier to reproduce, making it more feasible others will attempt to reproduce the claims made here.